# The Role of Stem Cells and Their Derived Extracellular Vesicles in Restoring Female and Male Fertility

**DOI:** 10.3390/cells10092460

**Published:** 2021-09-17

**Authors:** Ahmad Yar Qamar, Tariq Hussain, Muhammad Kamran Rafique, Seonggyu Bang, Bereket Molla Tanga, Gyeonghwan Seong, Xun Fang, Islam M. Saadeldin, Jongki Cho

**Affiliations:** 1College of Veterinary Medicine, Chungnam National University, Daejeon 34134, Korea; ahmad.qamar@uvas.edu.pk (A.Y.Q.); bangsk97@o.cun.ac.kr (S.B.); tanga@o.cnu.ac.kr (B.M.T.); 202050377@o.cnu.ac.kr (G.S.); fx2442@o.cnu.ac.kr (X.F.); islamms@cnu.ac.kr (I.M.S.); 2College of Veterinary and Animal Sciences, Jhang, Sub-Campus of University of Veterinary and Animal Sciences, Lahore 54000, Pakistan; tariq.hussain@uvas.edu.pk (T.H.); kamran.rafique@uvas.edu.pk (M.K.R.); 3Faculty of Veterinary Medicine, Hawassa University, Hawassa 05, Ethiopia

**Keywords:** infertility, fertility restoration, assisted reproductive technologies, stem cells, extracellular vesicles

## Abstract

Infertility is a globally recognized issue caused by different reproductive disorders. To date, various therapeutic approaches to restore fertility have been attempted including etiology-specific medication, hormonal therapies, surgical excisions, and assisted reproductive technologies. Although these approaches produce results, however, fertility restoration is not achieved in all cases. Advances in using stem cell (SC) therapy hold a great promise for treating infertile patients due to their abilities to self-renew, differentiate, and produce different paracrine factors to regenerate the damaged or injured cells and replenish the affected germ cells. Furthermore, SCs secrete extracellular vesicles (EVs) containing biologically active molecules including nucleic acids, lipids, and proteins. EVs are involved in various physiological and pathological processes and show promising non-cellular therapeutic uses to combat infertility. Several studies have indicated that SCs and/or their derived EVs transplantation plays a crucial role in the regeneration of different segments of the reproductive system, oocyte production, and initiation of sperm production. However, available evidence triggers the need to testify the efficacy of SC transplantation or EVs injection in resolving the infertility issues of the human population. In this review, we highlight the recent literature covering the issues of infertility in females and males, with a special focus on the possible treatments by stem cells or their derived EVs.

## 1. Introduction

Infertility is defined as an inability or failure to conceive or carry a pregnancy to term. Infertility is one of the most important issues affecting the health and social lives of individuals [1]. Both males and females equally contribute to the etiology of infertility, and its causes include age, anatomical defects, defective ovulation, male factors, infections, environmental issues, genetic disorders, and autoimmune diseases [2,3,4,5,6]. These factors may contribute to gamete production failure, reduced sperm concentration, fertilization failure, and implantation failure.

Scientists have adopted different approaches to overcome these issues and restore or enhance fertility. The choice of therapeutic approach is generally dependent on the etiology and duration of infertility, individual age, and personal preferences [6]. Available therapeutic approaches include hormonal therapy, medication for a specific etiology, surgical excision, fertility preservation, and assisted reproductive technologies (ART). Both hormonal and non-hormonal drugs may directly or indirectly induce sexual dysfunction and impair gamete production and maturation [7]. Therefore, cryopreservation of gametes is desirable if the use of reprotoxic drugs is necessary or if the effects of drugs on gametes are irreversible. Later, these cryopreserved gametes may be used through ART. Hormonal therapy is effective in certain cases of infertility such as defective ovulation and low sperm count; however, it is associated with other health concerns such as breast cancer [8,9]. Recent data indicate that ART can resolve almost 80% of infertility issues [10]. However, ART is highly invasive and is associated with health concerns such as hyperstimulation syndrome, which is caused by the use of hormones [11,12]. Furthermore, undesired multiple pregnancies, ethical concerns, and financial constraints reduce the effectiveness of ART [13].

## 2. Stem Cell for Therapy

Stem cells (SCs) are undifferentiated cells with the capability of self-renewal and differentiation into a wide range of cell types and are an integral part of the body’s internal repair system. SCs can divide essentially without limit, serve to replenish damaged cells [14,15], and produce highly differentiated descendants [16]. SC can be derived from adults, embryos, and cord blood sources [17]. Pluripotent SCs are categorized into embryonic and induced pluripotent stem cells that are generated by integrating programming factors into adult differentiated cells [18]. Moreover, mesenchymal SC populations are reported to be present in nearly all postnatal tissues and organs and are responsible for growth, homeostasis, and repair of aged or damaged tissue [19]; however, they exhibit a multipotent property. SCs work by secreting factors that actively regulate tissue regeneration including growth factors and cytokines [20]. Recently, scientists have begun to explore the therapeutic potential of SCs to treat different etiologies to restore fertility. Cell-based therapy is an avenue that may be utilized to take offspring from both infertile and sterile individuals [21]. Moreover, the isolation of SCs from the testis and endometrium has aided in gamete production in cases of sterility due to therapy. In a mouse model, SC transplantations have been found to be effective in restoring fertility [22,23]. In rhesus macaques, Hermann et al. reported the production of functional sperm following SC transplantation after chemically induced sterility [24]. Paradoxically, controversies and ethical concerns regarding stem cell therapy have been raised by many authorities. The efficiency of pluripotent stem cell therapy faces some drawbacks such as homing inefficiency, tumor formation, non-functionality, or immune rejection [25,26]. Furthermore, embryonic stem cells require the manipulation of gametes, early embryonic life, or induction of abortion, which interfere with some religious, political, and ethical guidelines [27]. Therefore, stem cell research requires some policies and ethical regulations to meet the challenges in their applicability for therapeutics [28].

## 3. Extracellular Vesicles as Alternative Therapeutics

Fortunately, SCs can secrete extracellular vesicles (EVs) that possess the same therapeutic and regenerative effects as the parent SCs [29]. EVs are nano-vesicles that contain proteins, lipids, RNAs (mRNA, miRNA, and lncRNA), and other biomolecules that play an imperative role in paracrine signaling [30,31]. EVs such as exosomes and microvesicles (MVs) are released from all living cells constantly. MVs are released from the cell membrane via direct budding, while exosomes are formed from multivesicular endosomes or multivesicular bodies [32]. Sorting of bioactive molecules such as RNA, miRNA, lipids, and metabolites is carried out through different ways of charging and loading into the EVs, as reviewed in [33]. Differences in EV size and/or surface compounds affect the capturing by the objective or acceptor cells. For instance, micropinocytosis is technically consistent with the capture of isolated exosomes and small EVs, but not large EVs or small EV aggregates [33,34]. Moreover, EVs can be internalized and targeted to the lysosome for degradation or recycled and re-released into the extracellular fluids. EVs can also transmit information to acceptor cells by local acting at the cell membrane, particularly for cell surface communication signals [35]. However, the main characteristic of EV cargo delivery is to fuse with the plasma membrane and transmit molecules into the cytoplasm [34] (Figure 1). The EVs are a snapshot of the secreting cells and contain the secretome of the cell of origin and can be used to treat different reproductive disorders due to their ability to transfer different molecules. EVs can be used for various cell-free therapeutics to overcome the drawbacks of stem cells that were presented previously [25,26].

Consequently, in this review, we discuss recent laboratory reports investigating cell-based and EV cell-free therapies used to treat infertility and restore fertility.

## 4. Stem Cells and Female Reproductive Issues

Infertility in females can be caused by various disorders such as Asherman syndrome (AS), adenomyosis, endometriosis, preeclampsia, premature ovarian failure (POF), polycystic ovary syndrome (PCOS), recurrent implantation failure (RIF), and tubular blockage. SC transplantation has been shown to improve ovarian function and reserves [36], initiate endometrial regeneration, restore endometrial function, and improve pregnancy outcomes [37]. In this section, we provide an insight into etiologies associated with female infertility and evidence of fertility restoration using SCs.

### 4.1. Premature Ovarian Failure

POF is a multifactorial disorder affecting females <40 years age and is characterized by amenorrhea, deficient ovarian steroids, increased gonadotropin levels, and infertility [38,39]. Developing follicles are usually absent; instead, ovaries have a network of connective tissue interspersed with fibroblasts. Moreover, estrogen (E_2_) deficiency results in the atrophy of the mucosa lining the uterus and vagina [40,41]. The exact cause of POF remains unclear; however, it is associated with chemo- or radiotherapy, smoking, metabolic disorders (classic galactosemia), viral infections (HIV and mumps), autoimmune disorders, and genetic disorders (such as fragile X syndrome, Turner’s syndrome, and inhibin alpha gene affection) [42]. No effective therapy has been reported to date. Hormonal therapy can only provide temporary relief from E_2_ deficiency. Alternatives such as egg donations are often intractable due to economic and ethical concerns [43].

Recently, SC transplantation has been recognized as an effective therapeutic tool for the restoration of ovarian function [44,45,46]. The idea of exploiting the therapeutic potential of SCs in mammalian females with POF is based either on the production of new oocytes from specialized germline SCs [47] or a reduction in the magnitude of cellular apoptosis [48]. SC transplantation enhances ovarian function in cases of POF through reduced apoptosis of granulosa cells with modulation of cytokine expression [49]. Similar results have been observed using SCs derived from different sources, such as adipose tissue [50,51,52], amniotic fluid [53], bone marrow [48,54], menstrual blood [55,56], umbilical cord [45,57,58,59], and skin [60]. Transplanted SCs migrate to the site of follicular damage and initiate repair [61]. Proper ovarian function is reflected by higher E_2_ levels and improved development of follicles, with an increased population of antral follicles, ultimately resulting in the establishment of a successful pregnancy [58]. The structural and functional integrity of ovarian tissue is restored through cellular differentiation [53,55], improved vascular remodeling [62], reduced apoptosis [48,61], and upregulated antioxidant factors [63]. For regeneration of damaged tissue, SCs actively release various factors, including fibroblast growth factor (FGF) [56], cytokeratin 8/18, proliferating cell nuclear antigen [57], transforming growth factor beta, and vascular endothelial growth factor [53]. Fouad et al. compared the therapeutic effects of human amniotic membrane-derived mesenchymal SCs (hAM-MSCs) and adipose tissue-derived MSCs (Ad-MSCs) in chemotherapy-induced ovarian insufficiency in rats [64]. Their results indicated that both cell types restored ovarian function; however, the therapeutic efficacy of hAM-MSCs was higher than that of Ad-MSCs.

Recent studies involved the use of SC-derived condition medium or EVs for treating patients with POF [65,66,67,68,69]. Cytokines present in these secretions may be responsible for angiogenesis, apoptosis, cell cycle, immuno-modulation, and recovery of ovarian function [69]. In animal models, SC-derived exosomes improved the follicular morphology, suppressed apoptosis [70,71], and restored the ovarian function [72] of POF patients. It was observed that these effects were mediated by different types of RNAs including miR-664-5p, miR-144-5p, and miR-1246. In another study, Ding et al. reported that SC-derived EVs can ameliorate POF through reduced oxidative stress (OS) [73]. EVs derived from BM-MSCs can protect the cells against the adverse effects of peroxide through the reduction of malondialdehyde and increased expression of superoxide dismutase 1 and catalase [74]. The reduction in the magnitude of OS may be due to the protective effect of EVs on mitochondria [74,75] (Table 1).

### 4.2. Polycystic Ovarian Syndrome

PCOS is a common reproductive problem that is associated with endocrine and metabolic disorders [76,77]. PCOS is characterized by anovulation along with a higher level of androgen and luteinizing hormone (LH), whereas follicle-stimulating hormone (FSH) either remains normal or low [78]. Patients with PCOS have menstrual dysfunction, abnormal hair growth, acne, alopecia, enhanced libido, and a higher percentage of miscarriage. Furthermore, PCOS is associated with a higher risk of coronary artery disease, diabetes (type II and gestational), endometrial carcinoma, hyperlipidemia, and stroke [79,80,81]. The exact cause of PCOS is unknown; however, it is suspected that it may be caused due to a defect in the hypothalamo-hypophyseal axis, abnormal ovarian steroidogenesis, and insulin resistance [78].

Stein and Leventhal recognized obesity as a predisposing factor responsible for PCOS [82]. Therefore, weight loss has been identified as an important treatment for PCOS patients. For hormonal disorders, oral contraceptives can be used to reduce serum androgen and LH levels resulting in a relief of acne and induction of regular menses [83]. Insulin sensitizers are used to overcome metabolic disorders that result in a reduced level of insulin. Recently, Xie et el. reported that MSCs have the potential to improve the histopathology and function of the ovaries affected with PCOS [84]. MSCs significantly reduce the gene expression of pro-inflammatory and fibrosis-related factors. Ultimately, the ovarian function was restored due to a reduction in inflammation. In another study, it was found that MSC-derived EVs can ameliorate the effects of PCOS through shuttle transfer of miR-323-3p that promoted the growth of cumulus cells and inhibited their apoptosis [85] (Table 1).

### 4.3. Endometrial Injuries

The endometrium is regenerated under the influence of ovarian E_2_ every month throughout the reproductive cycle [86,87]. The endometrium is also regenerated after parturition and endometrial resection. Zuo et al. reported the presence of an endogenous population of SCs in the uterus including endometrial epithelial SCs, endometrial MSCs, and endometrial endothelium SCs [88]. These multipotent cells expressed SC-specific surface markers [89,90] and have an imperative role in endometrial functioning. In animal models, endometrial-derived SCs can form endometrial tissues [89], and their regenerative potential is considered vital for the human endometrium during reproductive cyclicity [91].

Endometrial injuries lead to scar formation and loss of function, intrauterine adhesions, amenorrhea, and increased incidence of infertility or miscarriage [92,93]. In mice, intrauterine infusion of platelet-rich plasma (PRP) has been reported to enhance regeneration in damaged endometrium [94]. Platelets contain α-granules that secrete cytokines, chemokines, and growth factors [95]. These proteins have a paracrine effect on SCs that can initiate cellular migration, proliferation, and angiogenesis, and in turn induce cell regeneration.

MSCs have the potential to induce the repair of damaged endometrial stromal cells [96] and differentiate into endometrial epithelial cells [97]. MSCs induce the repair mechanisms through the release of factors including FGF, insulin-like growth factor-1, transforming growth factor-beta (*TGFβ1*), and vascular endothelial growth factor. In rats, acute endometrial injuries have been treated through transplantation of bone marrow-derived MSCs (BM-MSCs) [98]. MSCs migrated to the injured parts and secreted FGF, resulting in enhanced cellular proliferation of the endometrium and muscle layer. Moreover, MSC-derived growth factors can facilitate the regeneration of microvasculature, restore implantation, and support embryonic development. In another study, umbilical cord-derived MSCs (UC-MSCs) initiated the repair of damaged endometrium, resulting in the restoration of fertility [93]. Interestingly, miR-340 from EVs derived from bone marrow stem cells downregulated collagen 1α1, α-SMA, and *TGFβ1* in rats subjected to mechanical endometrial damage [99]. Furthermore, human umbilical cord mesenchymal stem cells derived EVs alleviated injured endometrial epithelial [100] and stromal cells [101] through increasing the Bcl-2 level and downregulating Cleaved Caspase-3 level, and it activated the PTEN/AKT signaling pathway to regulate proliferation and anti-apoptosis (Table 1).

### 4.4. Endometrial Atrophy

Endometrial atrophy is a rare condition in which the endometrial lining is not more than 5 mm in thickness [102]. Patients with this condition usually have poor reproductive outcomes. In many instances, the etiology of endometrial atrophy is unclear; however, prolonged use of oral contraceptives and tamoxifen risks this condition.

Chang et al. reported that patients with a thin endometrium (<7 mm) showed a satisfactory increase in endometrial growth after infusion with PRP, and they were able to achieve pregnancy [103]. Another research group observed similar outcomes using frozen embryo transfer [104]. Further investigations have demonstrated that PRP infusion promotes vascularization, as evidenced by increased vascular signals visible via the Doppler system [105].

SC therapy targeting the endometrial niche fills the cellular portion of the functional layers of the uterus. Santamaria et al. reported that SC therapy is an effective tool for the recovery of patients with endometrial atrophy [106]. Another study showed higher pregnancy and parturition rates using endometrium-derived MSCs (em-MSCs) in patients with thinned endometrium and with little or no responsiveness to the treatment with E_2_ [107]. Endometrial regeneration by SCs occurs via cellular differentiation and immunomodulation [108].

### 4.5. Repeated Implantation Failure and Recurrent Miscarriage

Repeated implantation failure (RIF) occurs when high-quality embryos produced through in vitro fertilization are repeatedly unable to implant [109]. Any abnormality of the embryo, endometrium, or immune system can lead to implantation failure. Successful implantation is dependent on the quality of the embryo, the implantation ability of the recipient endometrium, the maternal immune system [110,111,112], and paternal sperm factors. Maternal factors responsible for RIF include anatomical defects of the uterus, thrombophilia, diseases of connective tissue, endometrial thickness and non-receptivity, abnormal immune response [109], endometriosis [113], and competency of cumulus cells [114]. Embryonic factors responsible for RIF include genetic abnormalities and other intrinsic factors that impair embryonic development, hatching, and implantation. Treatment strategies should be dependent on the proper diagnosis of the factors responsible for RIF.

Recurrent miscarriage is defined as three consecutive pregnancy losses 20 weeks after the last menstruation. Causes of recurrent miscarriages include anatomical abnormalities of the uterus, antiphospholipid antibody syndrome, acquired or heritable thrombophilias, chromosomal abbreviations, environmental factors, infections, uncontrolled diabetes, unrecovered hypothyroidism, and other endocrine disorders [115,116].

Mammalian peripheral blood mononuclear cells (PBMCs), such as T- and B-lymphocytes and monocytes, exert a positive effect on the endometrium and its receptivity through cytokine secretion. Moreover, PBMCs help to establish the placenta and regulate immune tolerance during placentation [117]. Different research groups have reported that intrauterine application of PBMCs either alone [118], in medium supplemented with human chorionic gonadotropin [119], or co-cultured with luteal cells [120] resulted in significantly increased pregnancy, implantation, and live birth rates. Furthermore, Jensen et al. reported that fetal immuno-rejection can be prevented through the transfer of B-lymphocytes isolated from a normal gravid murine uterus to abortion-prone animals [121]. B-lymphocytes produce IgG-like antibodies that have a high affinity for antigens but are unable to trigger host defense mechanisms, thus protecting the fetus against maternally derived antibodies at the feto-maternal interface [122].

In addition to PBMCs, platelets are also involved in the implantation of human embryos [123]. Platelets play an imperative role in embryo–maternal communication and endometrial remodeling [124]. Moreover, platelets actively participate in corpus luteum formation by regulating neovascularization and luteinization [125]. Therefore, platelets may also help to increase the birth rate. This claim is well supported by the elevated pregnancy rate achieved following premating intrauterine infusion of PRP [126]. Similar outcomes have been observed in RIP-diagnosed patients treated with intrauterine administration of PRP [127].

Recently, SC therapy has been used to decrease abortion rates and improve reproductive outcomes. In mice, MSC transplantation resulted in improved reproductive performance through enhanced expression of interleukin 10 (*IL-10*) and *TGF-β* as well as reduced expression of tumor necrosis factor-alpha (*TNF-α*) and interferon-gamma (*IFN-**γ*) mRNA [128].

### 4.6. Asherman’s Syndrome

This is an acquired gynecological disorder characterized by intrauterine adhesions, fibrosis, hypomenorrhea, amenorrhea, and infertility [129,130]. AS is caused by endometrial destruction as a result of endometritis, repeated and aggressive curettages, and endometrial tuberculosis [130,131]. Intrauterine scar formation or adhesion impairs blastocyst implantation and leads to abnormal placentation, including placenta previa and accreta, and in turn this leads to recurrent miscarriage or infertility [129]. In AS, patients usually have reduced or no menstrual bleeding, pain, and infertility [132]. Hysteroscopic techniques for treatment have been developed but have inconsistent outcomes [106].

Recent research has shown the effective use of SCs in the treatment of AS. In humans, SCs from different sources including adipose tissue [133], bone marrow [106,134,135], and menstrual blood [136] have been used for treating endometrium affected by AS. Nagori et al. achieved the first successful conception in a patient with AS [135]. Intrauterine contraceptives were administered to this patient for 6 months without any improvement. However, endometrial regeneration was induced after implantation of endometrial angiogenic SCs isolated from adult autologous SCs. In 2014, another study was performed in patients with AS caused by endometrial tuberculosis [134]. Following transplantation of bone marrow-derived SCs, their menstrual flow became normal, but endometrial thickness did not increase beyond 6.7 mm, and patients failed to conceive. In a pilot cohort study in 2016, Santamaria et al. observed an immediate change in endometrial morphology after SC therapy [106]. CD133+ bone marrow-derived SC implantation resulted in enhanced endometrial thickness and neoangiogenesis. Moreover, resumption of menstruation and several pregnancies were achieved.

Recent studies have shown SC-derived EVs as a new therapeutic approach for treating animals with AS [99,137,138]. EVs worked by reducing fibrosis, promoting angiogenesis, and immuno-modulation. In rats with AS, BM-MSC-derived EVs promoted the growth of endometrial glands and decreased the magnitude of fibrosis [139]. It was suspected that endometrial repair was initiated by the EVs through inhibition of the *TGFβ_1_*/Smad2 pathway. In 2020, Saribas et al. reported that rats with damaged endometrium due to AS can be treated using exosomes [140]. It was observed that exosomes derived from endometrial MSCs caused a reduction in fibrosis and induction of cellular proliferation and vascularization in the uterus. EVs promoted angiogenesis due to the increased expression of vascular marker CD31 and VEGF receptor 1 (Table 1).

### 4.7. Endometriosis

Endometriosis is an E_2_-dependent inflammatory condition that involves the formation of endometrial stroma and glands outside the uterine cavity [142]. Endometrial tissue may be found around the ovaries, peritoneum, and uterosacral and broad ligaments. Patients with endometriosis suffer from chronic pelvic pain accompanied by menorrhagia, dysmenorrhea, dyspareunia, dyschezia, and dysuria [143]. Moreover, endometriosis is associated with a high risk of allergic disorders [144], autoimmune diseases [145], cardiovascular diseases [146], ovarian and endometrial cancers [147,148], and OS [149]. The exact mechanism underlying endometriosis remains unclear. However, different pathological conditions have been proposed as a cause of endometriosis, including retrograde menstruation [150], embryonic rest [151], coelomic metaplasia, implantation of endometrial SCs, dysregulation of the immune system, and lymphovascular spread [152,153]. Moreover, endometriosis may be due to endometrial-derived SCs [154] or too high mobilization of SCs from other sources of the body [155].

Previous therapies for endometriosis were based on treating infertility issues via hormonal therapy, removal of ectopic endometrial tissue by laparotomy, and pain management [98]. Infertility issues were treated through the induction of hypoestrogenism, amenorrhea, or endometrial atrophy [156] using nonsteroidal anti-inflammatory drugs, hormonal contraceptives, progestogens, anti-progestogens, agonists and antagonists of gonadotropin-releasing hormone (GnRH), ethyl estradiol, and aromatase inhibitors [153,157,158]. However, these hormonal therapies are inadequate, especially for patients with infertility due to suppressed ovarian function accompanied by endometrial atrophy [142].

The recent discovery of autologous em-MSCs has led to their possible use in the treatment of endometriosis. The main purpose of using SCs is to reduce inflammation after adhesiolysis. However, the use of SCs for treating endometriosis is controversial due to their possible role in the etiology of endometriosis. Gopalakrishnan et al. demonstrated the therapeutic potential of MSCs in devising a targeted anti-angiogenic therapy for patients suffering from endometriosis [159]. Endometriotic tissues have enhanced angiogenesis, so targeting angiogenesis can be helpful in curbing the progression of the disease. In the equine endometriosis model, MSC-derived EVs showed a prominent anti-inflammatory activity. EVs contain miRNAs that stimulate cell proliferation and reduce both the expression and secretion of pro-inflammatory factors [141] (Table 1). In another study, human Wharton’s jelly SC-derived conditioned medium and cell-free lysate inhibited endometriotic cells through the activation of intrinsic apoptotic pathways [160].

### 4.8. Adenomyosis

Adenomyosis is a condition in which the myometrial lining is invaded by endometrial tissue, resulting in the uterine enlargement that microscopically exhibits ectopic, non-neoplastic, endometrial glands and stroma surrounded by hypertrophic and hyperplastic myometrium. Endometrial tissue, including glands and stroma, is haphazardly located deep inside the myometrium, and many researchers think of adenomyosis (i.e., “internal endometriosis”) as a variant form of endometriosis [161]. Similar changes may be observed in the rectovaginal septum [162]. The etiology or developmental events that may be responsible for adenomyosis are usually unknown. The most widely accepted theory about the etiology of adenomyosis is that it may be due to down-growth and invagination of the basal layer of the endometrium into the myometrium [162,163]. Factors such as weak smooth muscles and increased intrauterine pressure, either alone or together, may also contribute to the occurrence of adenomyosis. Moreover, adenomyosis is maintained by high E_2_ levels and impaired immune system-associated growth regulation of ectopic endometrial tissue [162].

Adenomyosis is usually diagnosed in women aged 40–50 years, and it is occasionally diagnosed in young women who are either undergoing infertility evaluations or have symptoms of dysmenorrhea and menorrhagia. Adenomyosis can be diagnosed using ultrasonography, magnetic resonance imaging (MRI), and histological examination of biopsy samples. Adenomyosis is responsible for abnormal bleeding, pelvic pain, and infertility.

Adenomyosis is generally treated with the same drugs used to treat endometriosis [164]. Aromatase inhibitors, contraceptive pills, danazol, dimetriose, GnRH-analogues, prostaglandin antagonists, progesterone, and Chinese herbal products have all been used to control menstrual pain and menorrhagia in patients with adenomyosis [164,165]. Furthermore, adenomyosis can be treated using uterine artery embolization [166], excision of the myometrium or adenomyoma [167], myometrial electrocoagulation [168,169], and myometrial reduction [170]. Surgical excision is preferred if drugs fail to provide relief [164]. However, uncertainty in determining the exact site and extent of adenomyosis makes it difficult to accurately remove only the affected part of the uterus; therefore, hysterectomy is the most popular procedure [165]. Chen et al. isolated adenomyosis-derived MSCs (ADS-MSCs) with higher expression of cyclooxygenase-2 (*COX-2*) gene [171]. He suggested that *COX-2* plays a vital role in the pathogenesis of adenomyosis, as it may be responsible for controlling the migration, invasiveness, and proliferation of ADS-MSCs. Therefore, *COX-2* inhibition may be helpful in treating and preventing adenomyosis.

## 5. Stem Cells and Male Reproductive Issues

Male infertility is defined as the inability of a male to impregnate a fertile female. Male factor infertility is usually characterized by altered sperm motility, concentration, and morphology observed in samples collected 1 and 4 weeks apart [172]. Moreover, it may involve abnormal seminal volume and accessory sex gland function [173]. Male infertility may be caused by either congenital or acquired conditions, including genetic abnormalities, hormonal disturbances, immunological disorders, infection of the reproductive tract, testicular failure, systemic diseases, cancer, varicocele, altered lifestyle, and exposure to gonadotoxic factors [174]. However, in certain cases, the etiology remains unknown [175].

Among infertile males, 15–20% exhibit azoospermia, whereas 10% have a sperm count of less than 1 million/mL of semen [176]. Much effort has been focused on treating non-obstructive azoospermia, as this is the most challenging cause of male infertility and is difficult to treat [177]. Common medical treatments used for treating male infertility issues include surgical interventions, the use of hormones, and drug therapies. However, due to their limited therapeutic effect, there is a need to seek new treatment strategies. Currently, significant advances have been made in ART for treating male infertility. Semen cryopreservation along with intra-cytoplasmic sperm injection (ICSI) is the only technique used to preserve fertility. However, its low success rate [178], as well as infertility due to the absence of functional gametes, still pose a challenge for the current ART [179]. Following the discovery of SCs, scientists have proposed the use of SCs and their associated EVs as a new therapeutic approach to treat male infertility issues [180]. Recently, SCs and SC-derived vesicles have been reported to protect and restore the fertility of sperm during freeze–thaw procedures [181,182]. The identification and isolation of spermatogonial SCs (SSCs) have led to the possibility of generating gametes. Furthermore, the production of SSCs from BM-MSCs and Ad-MSCs both in vitro [183,184] and in vivo [185,186] can greatly facilitate fertility restoration in males. Therefore, SC transplantations in infertile males with normal genetic backgrounds may be considered a promising approach for fertility restoration [187]. In this section, the different causes of male infertility and the therapeutic potential of SCs to restore fertility are explained.

### 5.1. Hypogonadism

Male hypogonadism is a clinical syndrome caused by the low production of testosterone (T_4_), sperm, or both, and it can be caused by congenital or acquired disorders. Depending on etiology, hypogonadism is categorized as either primary or secondary. Primary hypogonadism is caused by testicular disease and results in low T_4_ concentration, impaired spermatogenesis, and high concentrations of gonadotropins. Secondary hypogonadism is caused by dysfunction of the hypothalamic–pituitary axis and results in low T_4_ concentration, reduced spermatogenesis, and low concentration of gonadotropins. In primary hypogonadism, spermatogenesis is impaired more than T_4_ production, whereas, in secondary hypogonadism, both functions are equally affected.

The treatment of hypogonadism is usually dependent on type. Primary hypogonadism does not respond to hormonal therapy because of the damaged seminiferous tubules; ART, donor sperm, and adoption should therefore be considered instead. In patients with secondary hypogonadism, fertility can be restored using GnRH or gonadotrophin therapy [188]. Furthermore, the use of human chorionic gonadotrophin or FSH in patients with hypogonadism can result in the enlargement of the testicle [189]. In mammals, FSH use can increase the population of SSCs [190,191]. The causes of hypogonadism and their treatments especially using SCs are explained as follows.

#### 5.1.1. Varicocele

Varicocele is an acquired cause of primary hypogonadism characterized by an abnormally enlarged collection of veins in the pampiniform plexus, which resembles a bag of worms [192]. It is the most common cause of male infertility and affects almost 15% of the population [193]. Varicocele is caused by impaired venous drainage that affects the countercurrent exchange mechanism of the testicles and results in elevated scrotal temperature and abnormal sperm production [194]. It may also result in impaired drainage of gonadotoxins from the testicles and testicular hypoxia [195]. The pathophysiology of varicocele is not well-defined; however, OS is the main factor responsible for abnormal sperm parameters. Patients with varicocele may be asymptomatic or may experience dull throbbing pain in the scrotum. Moreover, elevated temperatures may cause testicular atrophy due to the loss of germ cell mass [196].

Varicocelectomy is the first-line treatment; however, antioxidants may help to control OS [192]. In rats, berberine has been effective in varicocele treatment. Berberine resulted in improved self-renewal activities of SSCs and better Leydig–Sertoli cell communication through upregulation of BCL-6b, Etv5, and GDNF in SSCs [197]. SCs including SSCs and MSCs can inhibit apoptosis and necrosis; however, the exact antioxidant pathway is not clear [198].

#### 5.1.2. Testicular Trauma

Testicles are predisposed to traumatic injuries because of their external location. Testicular trauma should be treated as an emergency condition because exposure of the spermatic antigen to the immune system [199] may induce an auto-immune response against sperm through the production of anti-sperm antibodies (ASA) [200]. Among infertile males, 8–21% have ASA that can negatively affect sperm motility, capacitation, and acrosome [201]. Moreover, ASA can cause sperm lysis.

The current approach to treat testicular trauma involves surgical procedures including debridement and suturing to close the tunica albuginea and avoid exposure of spermatic antigen to the immune system. The auto-immune response can be prevented by closing the tunica albuginea within a maximum of 72 h post-trauma [202]. Previous reports have reported the failure of medications or orchiectomy to prevent an autoimmune response [203], and systemic use of steroids for the treatment of immunological infertility is restricted because of side effects.

Recent reports have demonstrated that MSCs have immunomodulatory effects and may control autoimmune diseases, including animal-specific sclerosis, diabetes, graft rejection, and rheumatoid arthritis [204,205,206]. In mice, BM-MSCs transfusion resulted in immunosuppression of ASA produced following traumatic rupture of the testicle [199].

#### 5.1.3. Testicular Torsion

Testicular torsion is considered an emergency condition [207] and can result in primary hypogonadism. Testicular torsion has a sudden onset, and patients experience intractable pain due to reduced blood supply to the testicles. Testicular torsion also results in germ cell injury, depending on the ischemia and the severity of the twist in the cord. In cases of testicular torsion, ischemia is usually followed by reperfusion, which in turn results in impaired spermatogenesis and the release of toxic substances such as free radicals from the damaged tissue into circulation [208,209,210]. During reperfusion, vascular endothelial injury and microcirculation disorders result in functionally compromised testicles.

Testicular torsion can lead to permanent dysfunction of the testicles; however, reduction and fixation of the spermatic cord within 6 h can significantly reduce its likelihood. In cases with a higher degree of twist, cellular necrosis occurs within 4 h, and a cord with a 360° twist can lead to complete or severe testicular atrophy within 24 h [211]. The recommended therapeutic approach is to ameliorate ischemic conditions, promote spermatogenesis, and regulate the immune response. Based on immunomodulatory effects, differentiation potential, and paracrine support, SCs have been reported to successfully treat torsion–induced infertility. SSCs injected into mice affected with testicular torsion restored fertility due to the upregulation of pre- and post-meiotic genes, which led to the improvement of testicular structure and function [212]. In another study, allogenic MSCs injected into torsion-induced infertile rats successfully restored fertility [207]. Hsiao et al. reported that fertility restoration in torsion-induced injury following MSC injections was the result of improved sperm motility and energy [213]. It was proposed that MSCs may be involved in the regulation of *Akt*/*GSK3* to create a balance between glycogenesis and glycolysis in sperm. Furthermore, Zhong et al. suggested that paracrine mechanisms of UC-MSCs can protect sperm against ischemic and reperfusion injuries by reducing cellular apoptosis, inflammation, and OS [214].

#### 5.1.4. Klinefelter’s Syndrome

Klinefelter’s syndrome (KS) is a congenital and common cause of primary hypogonadism. KS is a sex chromosome aneuploidy affecting 0.1–0.2% of newborn males. KS is characterized by small, firm testicles, androgen deficiency, azoospermia [215], and elevated gonadotropin levels [216]. The hormonal profile of KS patients is normal during their childhood, but germ cells are reduced due to the deterioration of testicles around mid-puberty [217]. KS patients had a low likelihood of fertility preservation before the introduction of ART, specifically ICSI [216]. A recent report described the extraction of microscopic sperm from almost 70% of KS patients aged 14–22 years [218]. In such patients, preserving testicular tissue containing SSCs before the onset of puberty may be helpful [219].

### 5.2. Epididymo-Orchitis

Epididymo-orchitis is inflammation of the epididymis and testicle, caused by infectious or non-infectious agents. Orchitis usually occurs as a result of epididymitis that spreads to the surrounding tissues [220]. Inflammation of the male genital organ affects fertility through reduced germ and Sertoli cell numbers [221]. Increased intratesticular pressure as a result of inflammation damages the germinal epithelium [194]. Different risk factors may favor epididymitis such as surgery or instrumentation of the urinary tract, prostatic obstruction, stenosis of urethral valves, long sitting hours, sexual activity, bicycle or motorcycle riding, and strenuous physical activity [222,223,224]. Based on the duration of symptoms, epididymo-orchitis can be classified as acute, subacute, or chronic [225]. The acute form persists for less than 6 weeks and is characterized by painful swelling. The chronic form persists for more than 3 months and is characterized by pain without swelling. Abscess, sepsis, and infertility are possible complications of epididymitis.

Treatment of epididymitis is usually based on the etiology. Different antibiotics, including ceftriaxone, doxycycline, azithromycin, and ofloxacin, have been reported to be effective against infectious agents [226,227]. Moreover, analgesic use, the elevation of the scrotum, reduced physical activity, and cold therapy reduces inflammation. In orchitis, supportive therapy along with the application of hot or cold packs and bed rest is recommended [225]. Antimicrobials should be avoided during viral orchitis. Erpenbach et al. and Ku et al. reported that INF-α can be used to prevent atrophy of testicles and infertility in the cases of bilateral orchitis caused by mumps [228,229]. However, Yeniyol et al. found that systemic use of INF cannot completely prevent testicular atrophy [230]. Recently, the transplantation of neonatal testis-cells combined with the use of antiviral drugs was found to be effective in restoring testicular tissue of mice with herpes virus induced-orchitis [221].

### 5.3. Gonadotoxicity

In males, reproductive toxicity is characterized by apoptosis of germ cells, azoospermia, and testicular atrophy [231,232]. Chemo- and radiotherapy used for treating cancer have detrimental effects on the gonads [233]. Moreover, gonadal health is negatively affected by alcohol, heavy metals, insecticides, pesticides, some antibiotics, tobacco, and recreational drugs [194]. There is a risk of permanent infertility following such treatments. In such circumstances, cryopreservation of sperm and immature testicular tissue containing SSCs is regarded as the gold standard for fertility preservation [234]. Gonadal tissue transplantation should only be considered when there is a low chance of retransmission of cancer cells. Therefore, the selection of SSCs using cell sorting techniques before transplantation is preferred [235].

In recent years, the therapeutic potential of SCs in gonadotoxicity has been investigated. MSCs have been reported to restore fertility in males with infertility induced by busulfan [186], cadmium [236], cisplatin [237], cyclophosphamide [238], and doxorubicin [239]. Similar findings were obtained in busulfan-induced azoospermic hamsters using intratubular injections of Ad-MSCs [185]. SCs can induce repair by reducing apoptosis, inflammation, and OS [237], and they can restore the normal morphology of epithelial cells lining the seminiferous tubules [186]. SCs can also reduce mitochondrial apoptosis to restore fertility [236]. Cai et al. reported that BM-MSCs restore fertility in cases of busulfan-induced azoospermia by releasing proteins, including angiogenic factors, antiapoptotic factors, chemokines, cytokines, and immunoregulators [240]. Furthermore, normal spermatogenesis was restored in the individuals with testicular toxicity using a conditioned medium derived from BM-MSCs [241] and exosomes derived from urine-derived SCs [242].

Very few recent studies have reported the use of stem cell-derived EVs for treating male infertility cases including the testicular gonadotoxicity and the non-obstructive azoospermia. We concluded these studies in Table 2. For example, testicular toxicity caused by chemotherapy or radiation was alleviated through using different kinds of stem cell-derived EVs [242,243,244,245]. Moreover, bone marrow mesenchymal stem cell-derived EVs alleviated the testicular ischemia-reperfusion injury in the rat model through reduction in apoptosis [246] (Table 2). Recently, we have reviewed the current clinical trials and animal models that use stem cells, or their derived EVs, and spermatogonial stem cells for the possible treatment of gonadotoxicity and the azoospermia cases [247].

## 6. Conclusions

Infertility is a global issue associated with various pathophysiological conditions. In the last few decades, the development of new therapeutic approaches for restoring fertility in infertile individuals has become an area of interest. The development of ART has led to improved treatments; however, certain cases remain untreatable. Moreover, ART is focused on preserving gametes rather than treating the source of infertility. Cellular therapy using SCs has led to alternative approaches to the treatment of infertility and restoration of fertility. Recently, cell-free therapy using stem cell-derived EVs provide a safe alternative to avoid the possible problems associated with SCs therapy. Several in vitro and in vivo studies have been performed to investigate the therapeutic potential of SCs and/or their derived EVs, which have produced satisfactory results. As most of these studies were performed on animal models, the effects of SCs and EVs therapy on the human population remain largely unknown. Hence, there is a need to further investigate the mechanisms of action, evaluate the side effects, establish quality control standards for clinical use, and legislate regulatory rules to ensure the safety of these promising therapeutic tools. We summarized the interventions and the uses of SCs and EVs for treating infertility issues in Figure 2.

## Figures and Tables

**Figure 1 cells-10-02460-f001:**
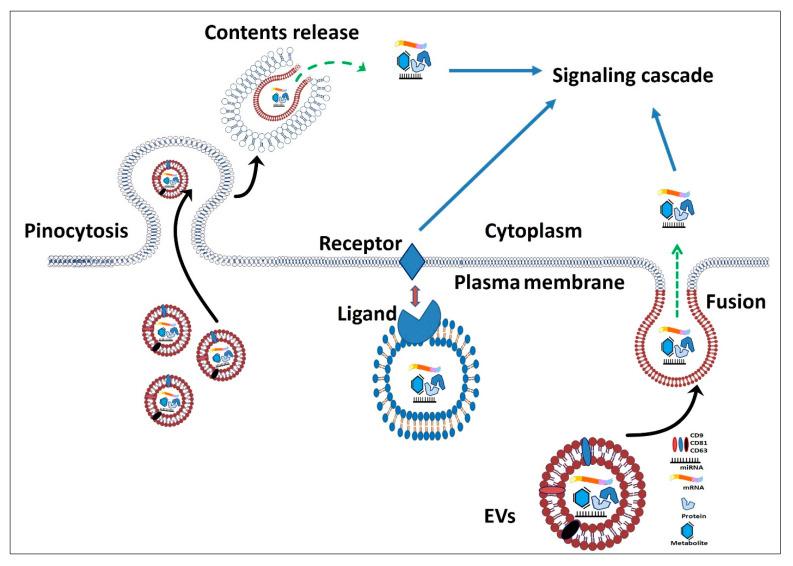
Schematic illustration of extracellular vesicles (EVs) uptake and cargo release to the target cells. EVs can fuse with the plasma membrane and deliver the molecular cargo into the cytoplasm. The EV surface protein can interact with cell receptors and deliver a signal pathway. EVs may be up-taken into the cells through pinocytosis where lysosomes can recycle or release their contents into the cytoplasm. The molecular contents of the EVs reflect the cell of origin and exhibit their actions when they become free from the lipid bilayer.

**Figure 2 cells-10-02460-f002:**
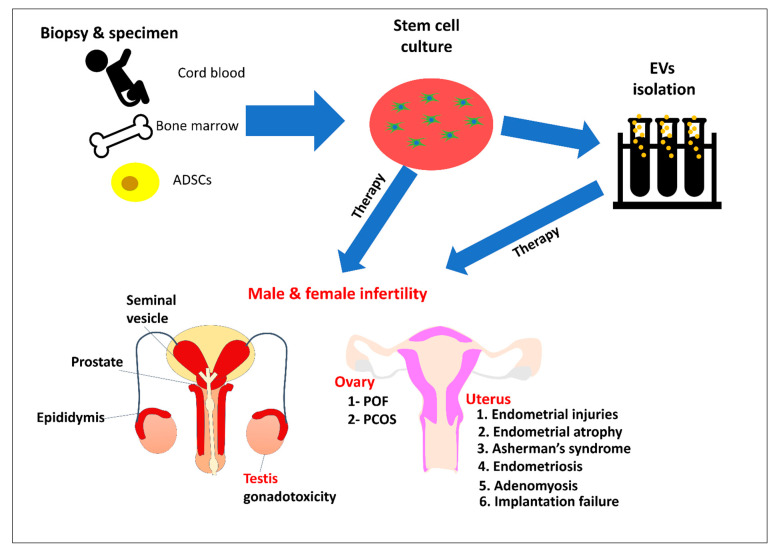
Diagrammatic illustration of the uses of stem cells (SCs) and their derived extracellular vesicles (EVs) for treating male and female infertility. Stem cells can be isolated from different sources such as cord blood, bone marrow, and adipose-derived stem cells (ADSCs) and used for culture and propagation. Cultured stem cells are used for isolating EVs from their conditioned medium. SCs and/or EVs can be administered to treat infertility diseases. POF: Premature ovarian failure. PCOS: Polycystic ovarian syndrome.

**Table 1 cells-10-02460-t001:** Uses of stem cell-derived extracellular vesicles for treating female infertility.

Female Infertility Category	Affection	Evs Source/Route of Administration in Vivo	Main Mechanism of Actions	Reference
Premature ovarian failure	Premature ovarian insufficiency (in vivo, mice model)	HUCMSCs/intravenous injection	Upregulation of AKT, p-AKT, and angiogenic cytokines (including VEGF, IGF, and angiogenin)	[65]
	Ovarian granulosa cell cisplatin-treated (in vitro, rats model)	HUCMSCs	Bcl-2 and caspase-3 were upregulated, whilst the expression of Bax, cleaved caspase-3, and cleaved PARP were downregulated	[66]
	Ovarian granulosa cell cisplatin-treated (in vitro, rats model)	HUCMSCs	Upregulated E2, StAR, and Bcl-2/Bax ratio downregulated Caspase 3	[68]
	Premature ovarian insufficiency (in vivo, mice model) & human granulosa-lutein cells (in vitro)	HAECs/intraovarian injection	109 cytokines involved in apoptosis, angiogenesis, cell cycle and immune response (in vivo) Decreasing apoptosis and TGF-β/Smad signaling pathway (in vitro)	[69]
	Premature ovarian insufficiency (in vivo, mice model)	BMSCs/intravenous injection	miR-664-5p targeted p53 and apoptosis	[70]
	Premature ovarian insufficiency (in vivo, rats model)	BMSCs/intraperitoneal injection	miR-144-5p targeted PTEN and apoptosis	[71]
	Premature ovarian insufficiency (in vivo, mice model)	HAECs/intraovarian injection	miR-1246 targeted phosphatidylinositol signaling and apoptosis pathways	[72]
	Premature ovarian insufficiency (in vivo, mice model)	AMSCs/intraovarian injection	miR-320a targeted SIRT4 and ROS formation	[73]
PCOS	Cumulus cells of PCOS patients (in vitro)	Human AMSCs	miR-323-3p promoted cell proliferation and inhibited apoptosis in CCs through targeting PDCD4	[85]
Endometrial injuries	Rats subjected to mechanical endometrial damage	BMSCs/intravenous injection	miR-340 downregulated collagen 1α1, α-SMA and transforming growth factor (TGF)-β1	[99]
	Mice injured endometrial epithelial cells (in vitro)	HUCMSCs	Inhibited IL-6, IL-1β, TLR4 and RelA Increased TNFA	[100]
	Human endometrial stromal cells injury (in vitro)	HUCMSCs	Increasing Bcl-2 level and downregulating Cleaved Caspase-3 level and activated the PTEN/AKT signaling pathway to regulate the proliferation and antiapoptosis	[101]
Asherman’s syndrome	Rat model (in vivo)	ADMSCs/intrauterine injection	Enhanced the expression of integrin-β3, LIF, and VEGF	[137]
	Rat model (in vivo)	HUCMSCs & collagen scaffold/endometrial transplantation	miRNAs facilitated CD163^+^ M2 macrophage polarization, reduced inflammation, and increased anti-inflammatory responses	[138]
	Rabbit model (in vivo)	BMSCs/Uterine muscle injection	CK19 level significantly increased whereas VIM level significantly decreased TGF-β1, TGF-β1R, and Smad2 mRNA were all significantly decreased	[139]
	Rat model (in vivo)	Uterus MSCs/intrauterine injection	MMP-2 and MMP-9 expression was enhanced TIMP-2 expression was decreased	[140]
Endometriosis	Lipopolysaccharides effects on endometrial cell (equines, in vitro)	Equine AMSCs	Reduced tumor necrosis factor-α (TNF-α), interleukin-6 (IL-6), interleukin 1β (IL-1β), and metalloproteinases (MMP) 1 and 13, and the release of some pro- or anti-inflammatory cytokines.	[141]

UCMSCs: human umbilical cord mesenchymal stem cells, HAECs: human amniotic epithelial cells, BMSC: bone marrow mesenchymal stem cell, AMSCs: amniotic mesenchymal stem cells, ADMSCs: Adipose mesenchymal stem cells.

**Table 2 cells-10-02460-t002:** Uses of stem cell-derived extracellular vesicles for treating male infertility caused by gonadotoxicity and non-obstructive azoospermia (NOA).

Model of Disease	Therapeutic Intervention/Route of Administration	Core Findings	Reference
Busulfan-induced NOA mice model	Urine-derived stem cells -derived EVs/Intratesticular	spermatogenic genes (*Pou5f1, Prm1, SYCP3,* and *DAZL*) and the spermatogenic protein UCHL1 were significantly increased after 36 days of injection	[242]
Busulfan-induced NOA rats model	Amniotic fluid-derived EVs/Intratesticular	DAZL and VASA were increased significantly. Sperm parameters and spermatogenesis index were significantly improved. OCT-3/4+ cells were increased in NOA rats after AF-Exos injection, showing the restoration of spermatogenesis.	[243]
Cyclophosphamide-induced testicular spermatogenic dysfunction	Bone marrow mesenchymal stem cell-derived EVs/intravenous	Increased spermatogonia cell proliferation and reduced apoptosis. Phosphorylated levels of ERK, AKT, and p38MAPK proteins were reduced	[244]
Electromagnetic field-induced oxidative stress in mouse spermatogonial stem cells (in vitro)	Sertoli cells-derived EVs	down-regulation of the apoptotic gene (Caspase-3), and oxidative stress. Up-regulation of SSCs specific gene (GFRα1).	[245]
Testicular ischemia-reperfusion injury in rats	Bone marrow mesenchymal stem cell-derived EVs/Intratesticular	Reduced HMGB1, caspase-3, and cleaved caspase-3	[246]

## Data Availability

The data supporting this study can be made available at reasonable request from the corresponding author.

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
