# Peer review of "The Role of Stem Cells and Their Derived Extracellular Vesicles in Restoring Female and Male Fertility"

_cells, 2021, doi:10.3390/cells10092460_

Round 1

Reviewer 1 Report

Summary and Abstract should be fused together.

A chapter on stem cells is missing.

Side effects of the stem-cell treatments should be listed as well as ethical issues relevant for the topics of the manuscript.

At least one scheme/chart should be added presenting the steps of collection/administration and mechanistic principles to gain beneficial effects of  stem-cell treatment for female and for male infertility (show cases). 

Author Response

  1. Summary and Abstract should be fused together.

R1. We thank the reviewer for this clarification. We fused the summary and abstract accordingly.

  1. A chapter on stem cells is missing.

R2. We added a concise review on this part.

  1. Side effects of the stem-cell treatments should be listed as well as ethical issues relevant for the topics of the manuscript.

R3. We added a concise chapter.

  1. At least one scheme/chart should be added presenting the steps of collection/administration and mechanistic principles to gain beneficial effects of  stem-cell treatment for female and for male infertility (show cases). 

R4. We thank the reviewer for the useful suggestion. We added Figure 2 to summarize the review.

Reviewer 2 Report

In their manuscript, Qamar et al systematically reviewed the role of stem cells and their derived extracellular vesicles in restoring fertility in males and females. The general topic is interesting and timely but requires revisions.

Too little was described EVs in the introduction, I would recommend describing the biogenesis (exosome, microvesicles, and apoptotic bodies) with the mechanism of action in the introduction (with illustrated figures). It would be better to have a dedicated section for EVs.

The uptake of EVs into cells is important for therapeutical functions. Please add all the ways of EV uptake into cells with figures and described them.

Table 1 & 2. I would suggest adding a route of administration of EVs into animal models.

Unfortunately, little was discussed about changes of therapies throughout the manuscript.

Where is the other perspective I would wish from a review?

Author Response

  1. In their manuscript, Qamar et al systematically reviewed the role of stem cells and their derived extracellular vesicles in restoring fertility in males and females. The general topic is interesting and timely but requires revisions.

R1. We thank the reviewer for the useful comments and suggestions to improve the quality of the manuscript. We have revised the manuscript accordingly.

  1. Too little was described EVs in the introduction, I would recommend describing the biogenesis (exosome, microvesicles, and apoptotic bodies) with the mechanism of action in the introduction (with illustrated figures). It would be better to have a dedicated section for EVs.

R2. We thank the reviewer for the suggestion. We added a concise paragraph on this subject.

  1. The uptake of EVs into cells is important for therapeutical functions. Please add all the ways of EV uptake into cells with figures and described them.

R3. We thank the reviewer for the suggestion. We added a concise paragraph and a figure on this subject.

  1. Table 1 & 2. I would suggest adding a route of administration of EVs into animal models.

R4. We thank the reviewer for the suggestion. The route of administration has been added in the appropriate in vivo experiments.

  1. Unfortunately, little was discussed about changes of therapies throughout the manuscript.

R5. In our review, and based on the ethical and problematic issues of using stem cell therapy, we focus on the alternatives to stem cell therapy for treating infertility, to highlight the important roles of extracellular vesicles in this regards.

  1. Where is the other perspective I would wish from a review?

R6. We edited the manuscript for clarification of the objectives.

Round 2

Reviewer 1 Report

I find the revised manuscript acceptable.

Reviewer 2 Report

Concerns and suggestions were addressed.